# Dynamic predictors of COVID-19 vaccination uptake and their interconnections over two years in Hong Kong

Jiehu Yuan [1,7], Yucan Xu[1,7], Irene Oi Ling Wong[2], Wendy Wing Tak Lam[1,3], Michael Y. Ni [1,4,5], Benjamin J. Cowling [2,6] ✉ & Qiuyan Liao [1] ✉

The global rollout of COVID-19 vaccines faces a significant barrier in the form of vaccine hesitancy. This study adopts a dynamic and network perspective to explore the determinants of COVID-19 vaccine uptake in Hong Kong, focusing on multi-level determinants and their interconnections. Following the framework proposed by the Strategic Advisory Group of Experts (SAGE), the study used repeated cross-sectional surveys to map these determinants at multiple levels and investigates their interconnections simultaneously in a sample of 15,179 over two years. The results highlight the dynamic nature of COVID-19 vaccine hesitancy in an evolving pandemic. The findings suggest that vaccine confidence attitudes play crucial roles in vaccination uptake, with their importance shifting over time. The initial emphasis on vaccine safety gradually transitioned to heightened consideration of vaccine effectiveness at a later stage. The study also highlights the impact of chronic condition, age, COVID-19 case numbers, and non-pharmaceutical preventive behaviours on vaccine uptake. Higher educational attainment and being married were associated with primary and booster vaccine uptake and it may be possible to leverage these groups as early innovation adopters. Trust in government acts as a crucial bridging factor linking various variables in the networks with vaccine confidence attitudes, which subsequently closely linked to vaccine uptake. This study provides insights for designing future effective vaccination programmes for changing circumstances.

In January 2020, the World Health Organisation (WHO) declared coronavirus disease 19 (COVID-19) a pandemic. Since then, the world experienced numerous epidemic waves. Around 7 million deaths have been reported as of May 2023 when the WHO declared that COVID-19 "no longer constitutes a public health emergency"[1]. The true death toll is even higher. COVID-19 vaccines were developed at record speed in response to the emergence of COVID-19[2], with vaccination campaigns having been rolled out worldwide since late 2020 and early 2021. However, despite strong evidence on the effectiveness of current COVID-19 vaccines for preventing severe illness, hospitalisation, and deaths associated with COVID-19[3–5], vaccine hesitancy remains widespread and an important barrier to high population uptake of the vaccines[6].

[1]School of Public Health, Li Ka Shing Faculty of Medicine, The University of Hong Kong, Hong Kong, China. [2]WHO Collaborating Centre for Infectious Disease Epidemiology and Control, School of Public Health, Li Ka Shing Faculty of Medicine, The University of Hong Kong, Hong Kong, China. [3]Li Ka Shing Faculty of Medicine, Jocky Club Institute of Cancer Care, The University of Hong Kong, Hong Kong, China. [4]State Key Laboratory of Brain and Cognitive Sciences, The University of Hong Kong, Hong Kong, China. [5]Urban Systems Institute, The University of Hong Kong, Hong Kong, China. [6]Laboratory of Data Discovery for Health Limited, Hong Kong Science and Technology Park, New Territories, Hong Kong, China. [7]These authors contributed equally: Jiehu Yuan, Yucan Xu. ✉e-mail: bcowling@hku.hk; qyliao11@hku.hk

COVID-19 vaccine hesitancy is complicated and shaped by an array of psychosocial and contextual factors as well as their interactions[7–10]. The WHO Strategic Advisory Group of Experts (SAGE) on Immunisation Working Group proposed a framework to map determinants of vaccine hesitancy into three main categories including individual and interpersonal factors, vaccine/vaccination specific factors, and contextual factors[11]. Individual factors included demographics, perceived risk of the pandemic[7,12], adoption of alternative preventive behavioural patterns[13,14], perceived self-efficacy in preventing the infection[7,12], trust in government[15,16], physical health status[14,17,18], and psychological distress[18–20]. Vaccination can be motivated by self-protection or protecting the loved ones[21]. Hence, interpersonal factors indicating by one's co-habiting characteristics[17,22] are also important determinants of vaccine acceptance. Vaccine/vaccination specific factors refer to vaccine safety, vaccine side effects, vaccine efficacy, and vaccine delivery and administration such as price, accessibility, vaccine pass and mode of administration[19,23,24]. Vaccine hesitancy is specific to contexts. At the contextual level, the evolving pandemic situation (i.e., reported cases and death numbers)[25], one's living neighbourhood (indicating the influence of personal socioeconomic status)[20,26,27], and the widespread misinformation from social media[28] can also shape people's vaccine hesitancy.

While mapping the multilevel determinants of vaccine hesitancy or uptake simultaneously can depict a comprehensive picture for guiding the design of vaccination programme[21], existing studies on COVID-19 vaccine hesitancy or acceptance mainly included individual/interpersonal and vaccine-specific factors[29–31]. Contextual determinants of vaccination uptake received increasing attention but were usually studied separately with other levels of determinants[15,20,27]. It is suggested that individual and vaccine-specific factors are interconnected to co-shape people's vaccination decision[32]. There remains limited understanding about how the contextual factors interact with individual and vaccine-specific factors. In this study, following the framework proposed by WHO SAGE on Immunisation Working Group[11], we adopted a network approach to map individual, interpersonal, vaccine-specific and contextual determinants of vaccine hesitancy simultaneously[33]. Compared with conventional statistical models such as multivariable (linear or logistic) regression models, a mixed graphical network approach has the strengths of incorporating various data formats and allowing mutual interactions amongst variables and hence can reveal complex pathways lying between multilevel factors and vaccine uptake[33].

Vaccine hesitancy is also dynamic, changing with the evolving situation[13]. First, public perceptions of the disease risk versus vaccine risk can change due to the rapid evolution of the pandemic situation and policy response[34]. Second, media focus may shift from disease risk to vaccine risk as disease incidence in the community declines and vaccines become widely available[35]. Third, people's experiences with the new vaccines also evolve as more people initiate their vaccination, which in turn reshapes their attitudes toward the vaccines and intention to take a second or third dose of the vaccines[36,37]. Furthermore, policies or official recommendations for COVID-19 vaccination have been constantly changed as more evidence on the characteristics of diseases and the efficacy of vaccines became available[38]. One key change lies in the shift of the initial recommendation for completing two doses of COVID-19 vaccines (the primary doses) to the uptake of a third dose (booster dose) 6 months after receiving the primary doses[39] due to observing a decline in neutralising antibodies among vaccinated individuals[40]. Strategies for promoting the uptake of the primary and booster vaccinations should be different[41], highlighting the importance of a tailored approach for vaccination promotion by stage of the vaccination campaign. Overall, it indicates that it is essential to monitor and track the dynamics of determinants of vaccine uptake throughout the vaccination campaign. A few large-scale population-based studies have been conducted to tackle the temporal changes of determinants on vaccination. These studies concluded that the key determinants of vaccine uptake varied by stage of the pandemic and vaccination campaign[6,7,19]. However, these studies only covered a short rather than the whole period (from the initial rollout to a scale-up phase) of the COVID-19 vaccination programme.

Hong Kong represents an excellent case study for reviewing the vaccination campaign in the context of high vaccine accessibility but low vaccine acceptability[42]. Hong Kong launched its vaccination programme by offering two types of free vaccines for its people: the mRNA vaccine BNT162b2 (BioNTech/Fosun Pharma/Pfizer) and the inactivated vaccine CoronaVac (Sinovac), since February 2021[43]. Sufficient vaccines for the whole population of over 7 million were procured and made free and easily accessible for the population by setting up multiple vaccination centres in the community[44]. Therefore, vaccine accessibility was not considered to be a major barrier to vaccination uptake in this study. However, public confidence in the vaccines was greatly dampened by various negative vaccine-related news such as sudden deaths immediately following COVID-19 vaccination[45]. Various strategies have been employed to boost the population's vaccination uptake in Hong Kong. Three months after the vaccine rollout, the government invited the business sectors and social organisations to provide various incentives to encourage vaccine uptake in the population, including materials gifts, financial incentives, and vaccination leaves[46]. In view of the waning immunity and the emergence of new virus variants, the Hong Kong government started to promote a third dose booster uptake in high-risk groups on 3 November 2021. The booster recommendation was expanded to the general population who finished their primary doses uptake for at least six months on 23 November 2021[44]. However, by 23 December 2021, shortly before Omicron was introduced into the community, only 49% of persons aged 60 years or above had received at least two doses of COVID-19 vaccination in Hong Kong and only 7% had received a third dose. The low vaccination uptake in Hong Kong resulted in more than 200 deaths per day during the explosive epidemic wave caused by the Omicron BA.2 variant from February to April 2022, pushing the city's daily COVID-19 mortality rate to the highest in the world[47]. On 21 February 2022, the Hong Kong government started to implement its most stringent vaccine mandatory arrangement (or "Vaccine Pass"), requiring all eligible persons to receive at least one dose of COVID-19 vaccine to gain access to specified premises. The vaccine mandate was later expanded to cover children aged 5–11 years on 30 September 2022[44]. The chronology of pandemic waves and major vaccination programme events are shown in Fig. 1

The catastrophic impact of COVID-19 pandemic has revealed flaws in the vaccination programme, while some were locally relevant, most were more systematic and ubiquitous across jurisdictions. A successful vaccination programme requires timely and responsive understanding of vaccination determinants. We aimed to answer a crucial yet under-investigated research question: What are the key determinants and their interactions associated with the primary and booster vaccination uptake at different stages of the COVID-19 vaccination campaign? Drawing experiences from Hong Kong, we aimed to inform more efficient vaccination programmes that are relevant for the international communities. We sampled a large, representative sample in successive epidemic waves throughout the pandemic to systematically compare important determinants of vaccine uptake at different stages. A comprehensive and identical set of individual psychosocial and contextual variables which can be mapped onto three levels were included: individual/interpersonal level (i.e., COVID-19 risk perceptions, trust in government, physical and mental health status, and demographics), contextual level (i.e., living community vulnerability level, pandemic-related situation changes such as COVID-19 report cases and death numbers), and vaccine/vaccination specific level (i.e., COVID-19 vaccine confidence) to assess which are more important for people's vaccine uptake. We used a mixed graphical network model to depict the multiple and complex interconnections between all these variables within a

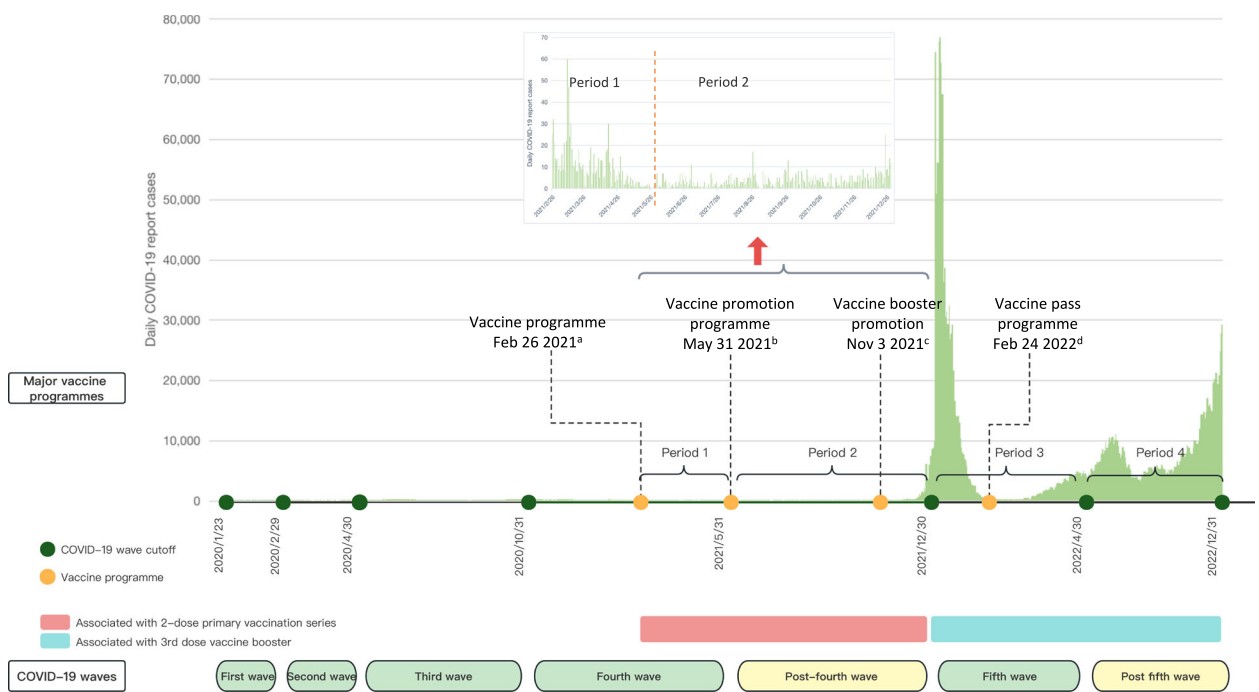

**Fig. 1 | COVID-19 pandemic waves, survey periods and major policies in Hong Kong.** There are generally five pandemic waves and two post-wave periods up to the end of 2022. The vaccine programme in Hong Kong started in February 2021, when we began collecting data on COVID-19 vaccination intention and uptake from the public on a regular basis. We partitioned the survey periods according to the pandemic wave to investigate the dynamic changes of the COVID-19 pandemic and public vaccine acceptance. **a** Period 1 covers the period from the initiation of the vaccine programme to the end of the fourth epidemic wave. Priority vaccinations were first arranged for five high-risk groups since the end of February 2021. The vaccine eligibility was expanded to cover adults aged 30 or above since 15 March 2021, and was further expanded to cover all residents aged 16 or above on 15 April 2021. **b** Period 2 corresponds to the post-fourth wave period, where the only

cases detected were in inbound travellers and which largely overlapped with the vaccine promotion programme featuring various incentives to boost vaccination. **c** Period 3 corresponds to the fifth wave, during which vaccine booster promotion and a mandatory vaccine pass scheme were implemented. The Hong Kong government started to promote a third dose booster uptake in high-risk groups on 3 November 2021. The booster recommendation was expanded to the general population who finished their primary doses uptake for at least six months on 23 November 2021. **d** Period 4 represents the post-fifth wave when the vaccine pass scheme included progressively stricter requirements and greater restrictions for the unvaccinated. The outcome in Periods 1 and 2 was the 2-dose primary vaccination series, while the outcome in Periods 3 and 4 was the third dose vaccine booster.

system. We also examined textual responses to depict people's vaccine hesitancy reasons at a later phase of the vaccine rollout (11 months after the vaccination programme had been launched).

## Results

### Study timeline

As of 31 December 2022, Hong Kong has experienced five pandemic waves. The first four waves were milder, resulting in less than 1% of the population being infected, while the fifth one was severe with high morbidity and mortality (Fig. 1). The rollout of the vaccination programme started in the middle of the fourth pandemic wave. Our study timeline covered a period from the initial rollout of the COVID-19 vaccination programme to a break period following the waning of the fifth wave of the pandemic in Hong Kong, a period when Hong Kong gradually entered the post-pandemic era[44]. Repeated cross-sectional telephone surveys were successively conducted (at weekly/bi-weekly/monthly interval) during our study timeline. Based on the timeline of the pandemic waves, we combined and divided our surveys into four periods: a period during the fourth pandemic wave (P1: Feb 22 – May 31, 2021), a post-fourth wave period (P2: June 1–Dec 30, 2021), a period during the fifth wave (Omicron wave) (P3: Dec 31, 2021– Apr 30, 2022), and a post-fifth wave period (P4: May 1–Dec 30, 2022) (Fig. 1). The division of the four survey periods was aligned with the major stages of COVID-19 vaccination campaign in Hong Kong: P1 covers an initial period of vaccine rollout before the introduction of various incentive strategies; P2 represents a scale-up phase when various incentives were introduced to boost vaccination uptake, with promotion

focusing on completion of the primary doses uptake (the first two vaccine doses); P3 is a period when the vaccine pass policy was announced and implemented to promote uptake of the third dose booster; and P4 represents a scale-up phase with progressively stringent requirements and specific penalties were introduced for children and adults who had received fewer than three vaccine doses.

### Participants' characteristics and vaccine uptake across the four periods

Across P1-P4, a total of 15,179 participants completed the surveys and were included in data analyses. Distribution of participants' gender, age, educational attainment, and employment status were comparable with the most recent census data based on effect size, except for a higher proportion of female and older participants in P4 (Table 1). In P1, of 3523 participants, only 14.8% (95%CI: 13.7–16.0%) reported that they had received the first dose of COVID-19 vaccines. In P2, of 7056 participants, the first-dose completion rate increased to 64.6% (95%CI: 63.5–65.7%), four to 11 months after the vaccine rollout. The second dose completion rate was 56.8% (95%CI: 55.6–58.0%), slightly lower than the uptake rate of the first dose. By P2, the booster-completion rate remained low (3.2%, 95%CI: 2.5–4.0%). In P3, of 2580 participants, 22.4% (95%CI: 20.8–24.1%) reported that they had received the booster dose while the uptake rate of the two primary doses reached 77.5% (95%CI: 75.9–79.1%). In P4 of 2020 participants, the booster-completion rate increased to 75.1% (95%CI: 73.2–77.1%), while the primary-doses completion rates reached 92.9% (95%CI: 91.7–94.0%). Temporal changes of the first-, second-, and third-dose vaccine uptake

**Table 1 | Participants' demographics comparing with population distribution using census data**

| Demographic characteristics | P1 (Feb 22–May 28 2021) (%) | P2 (Jun 21–Dec 16 2021) (%) | P3 (Jan 3–Mar 10 2022) (%) | P4 (Jun 6–Nov 17 2022) (%) | Total sample (%) | Population distribution (%)[a] |
|---|---|---|---|---|---|---|
| Sample size (N) | 3523 | 7056 | 2580 | 2020 | 15,179 | |
| Gender[b] | | | | | | |
| Female | 57.6 | 60.4 | 57.2 | 64.8 [c] | 59.8 | 52.9 |
| Male | 42.4 | 39.6 | 42.8 | 35.2 [c] | 40.2 | 47.1 |
| Age group (years) | | | | | | |
| 18–24 | 8.0 | 6.5 | 5.5 | 4.8[c] | 6.5 | 6.9 |
| 25–34 | 13.4 | 11.9 | 11.4 | 11.7[c] | 12.1 | 14.6 |
| 35–44 | 14.5 | 13.7 | 15.4 | 12.8[c] | 14.1 | 16.7 |
| 45–54 | 14.8 | 15.8 | 16.1 | 14.0[c] | 15.4 | 18.0 |
| 55–64 | 16.8 | 16.6 | 16.4 | 16.1[c] | 16.6 | 20.1 |
| 65 or above | 30.5 | 33.1 | 33.1 | 38.3[c] | 33.2 | 23.7 |
| Educational attainment | | | | | | |
| ≤Primary | 16.9 | 18.2 | 17.1 | 20.9 | 18.1 | 18.9 |
| Secondary | 44.0 | 43.5 | 46.0 | 43.7 | 44.1 | 45.9 |
| ≥Tertiary | 37.8 | 36.6 | 35.6 | 34.1 | 36.3 | 35.2 |
| Employment status | | | | | | |
| Employed/Students/home makers/retirees | 94.6 | 94.6 | 94.5 | 95.4 | 94.7 | 90.0 |
| Unemployed[d] | 3.9 | 3.7 | 4.2 | 3.1 | 3.8 | 2.4 |

[a]Population data were obtained from the 2021 Hong Kong census data.

[b]Self-reported gender.

[c]Effect size equal or greater than 0.30 compared with the population census data. Effect size was calculated with the formula $\omega = \sqrt{\sum_{i=1}^{m} \frac{(p_0(i)-p_1(i))^2}{p_0(i)}}$ where $p_0$ and $p_1$ are the observed proportions in the $i$th category of the census data and survey data, respectively. An effect size of ≥0.30 indicating a moderate-to-above differences between the sample and the population structure in terms of demographic distributions.

[d]Unemployment group included unemployed persons or who reported that not working for other reasons.

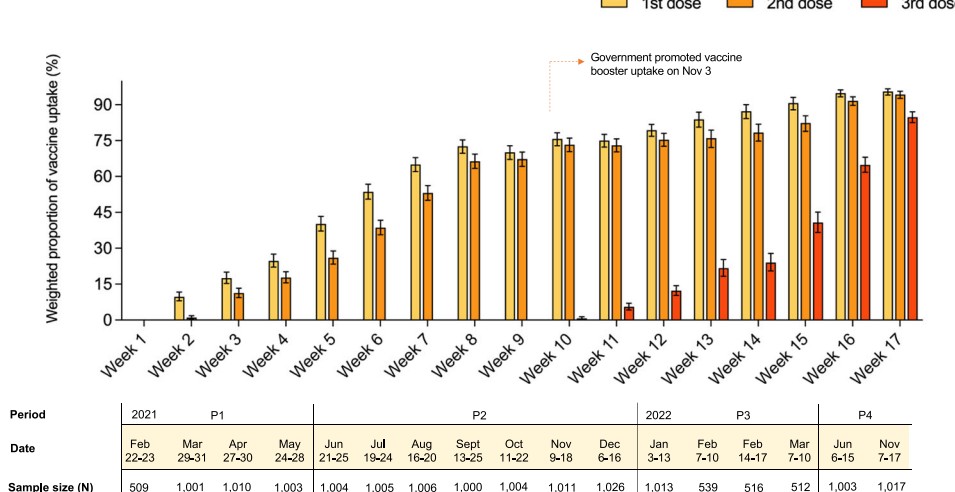

**Fig. 2 | Weighted proportion of vaccine uptake by number of doses completed across 17 survey weeks and the four divided periods of the study ($N$ = 15,179).** Bar graph illustrating the proportions of the first dose/second dose/third dose of vaccine uptake, weighted to the adult population based on Hong Kong census data in 2021, with 95% confidence interval ± about 3%. 95%CI was calculated with the normal approximation. The below horizontal bar shows the period, specific dates, and sample size for each survey wave.

rates amongst surveyed participants throughout the study periods are provided in Fig. 2. We also provided vaccine uptake rates by participants' demographics and their self-reported level of trust in government in Supplementary Table 1.

## Dynamic changes of determinants associated with vaccine uptake through a network lens

We included a set of identical determinants that were potentially associated with vaccination uptake to estimate the network in each

period to optimise the comparability of networks across P1–P4 (Fig. 3). Overall, the network explained 10.1% in P1, 18.2% in P2, 9.5% in P3 and 12.6% in P4, variance of the vaccine uptake. The predictability was comparable across networks, with that for P2 being slightly higher than those in other periods, indicating that the determinants included in the network can explain a greater proportion of variance in the vaccine uptake in P2. As is shown in Fig. 3, several major patterns can be identified by comparing the networks across P1–P4. First, vaccine confidence variables consistently clustered together and linked closely to

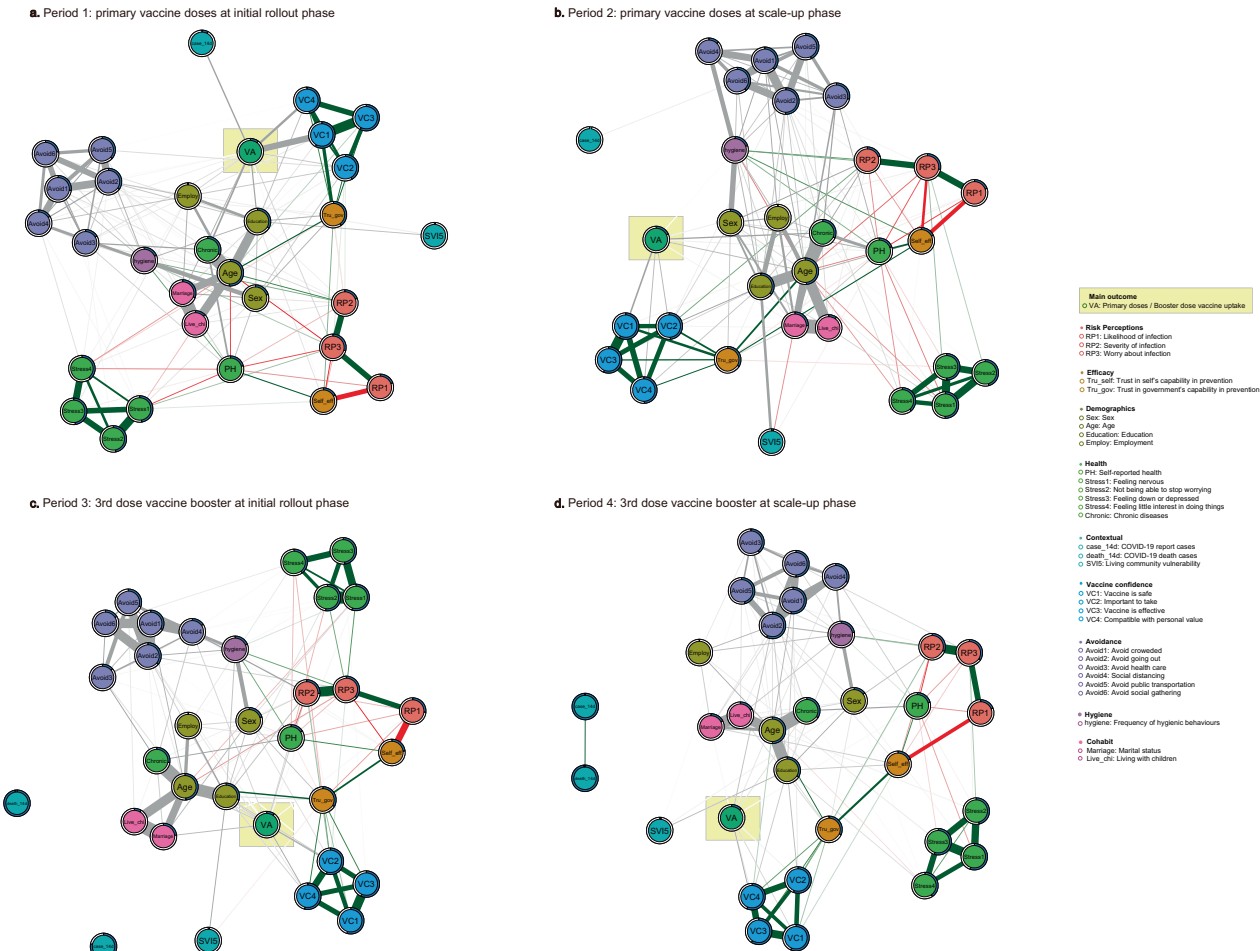

**Fig. 3 | Networks of determinants associated with vaccine uptake across P1-P4.** All the variables input in the network across the four periods remained the same, except that number of deaths was included in P3 and P4 but not P1 or P2 due to the extremely low number of human deaths due to COVID-19 in either period. The outcome variable (vaccine uptake) was highlighted in yellow square in each panel figure to ease interpretation. **a–d** show the conditional dependency amongst all the nodes within the network from P1 to P4. We manually categorised all the independent variables into nine domains, as shown with different node colours. Edge thickness indicates the magnitude of the partial correlation between nodes. Green edge represents positive association between continuous variables, red edge represents negative association between continuous variables. No signs were assigned to interaction involving categorical variables, thereby they presented as grey colour. The grey circle surrounding the node indicates the predictability of the node by other nodes.

vaccine uptake across P1–P4, representing the most proximal determinants associated with both the uptake of the primary vaccine doses and the booster dose. Second, demographic variables exhibited great proximity to the vaccine uptake, with chronic disease status and age among all included demographic variables having the strongest edges linking to vaccine uptake across P1–P4. Third, the contextual factors, including the social vulnerability of participants' residential community, the number of reported cases 2 weeks before the survey date (hereafter we termed it as number of cases), and the number of deaths 2 weeks before the survey date (hereafter we termed it as number of deaths), were generally independent of vaccine uptake, with only the number of cases being associated with the uptake of primary vaccine doses in P1. Fourth, the edges between non-pharmaceutical preventive behaviours and vaccine uptake persistently existed. However, risk perceptions of COVID-19 and psychological distress were placed more peripherally in the networks across P1–P4, suggesting that these variables were not central for determining vaccine uptake.

### Relative importance of determinants of vaccine uptake across the four waves

We further extracted the relative importance (pairwise edge weights) of all the variables to have a closer look at the exact nature of interactions between all the variables and vaccine uptake across P1–P4. The relative importance of continuous variables was directly extracted from the weighted adjacency matrix. While for categorical variables that include more than one level (i.e., one parameter for each of the two or more categorical levels), the edge weight was averaged by the absolute value of all parameters. The dynamic changes of the relative importance of determinants associated with vaccination uptake are shown in Fig. 4. We input the vaccine confidence variables into the network models separately rather than feeding the model with an aggregated score, which allows us to have a more nuanced understanding on which attitudinal variable is more important for vaccine uptake across P1–P4. It shows that vaccine safety was more important at the beginning of the vaccination programme, but its relative importance declined in the later stages of the vaccination campaign. The relative importance of vaccine effectiveness showed an opposite trend. Vaccine effectiveness had no significant association with vaccine uptake at the initial phase of vaccine rollout but became a key determinant in P3 and the most important determinant in P4, associated with uptake of the booster dose.

Amongst demographic variables, chronic disease status remained one of the most important barriers to both uptake of the primary doses and the booster dose, but its relative importance gradually declined

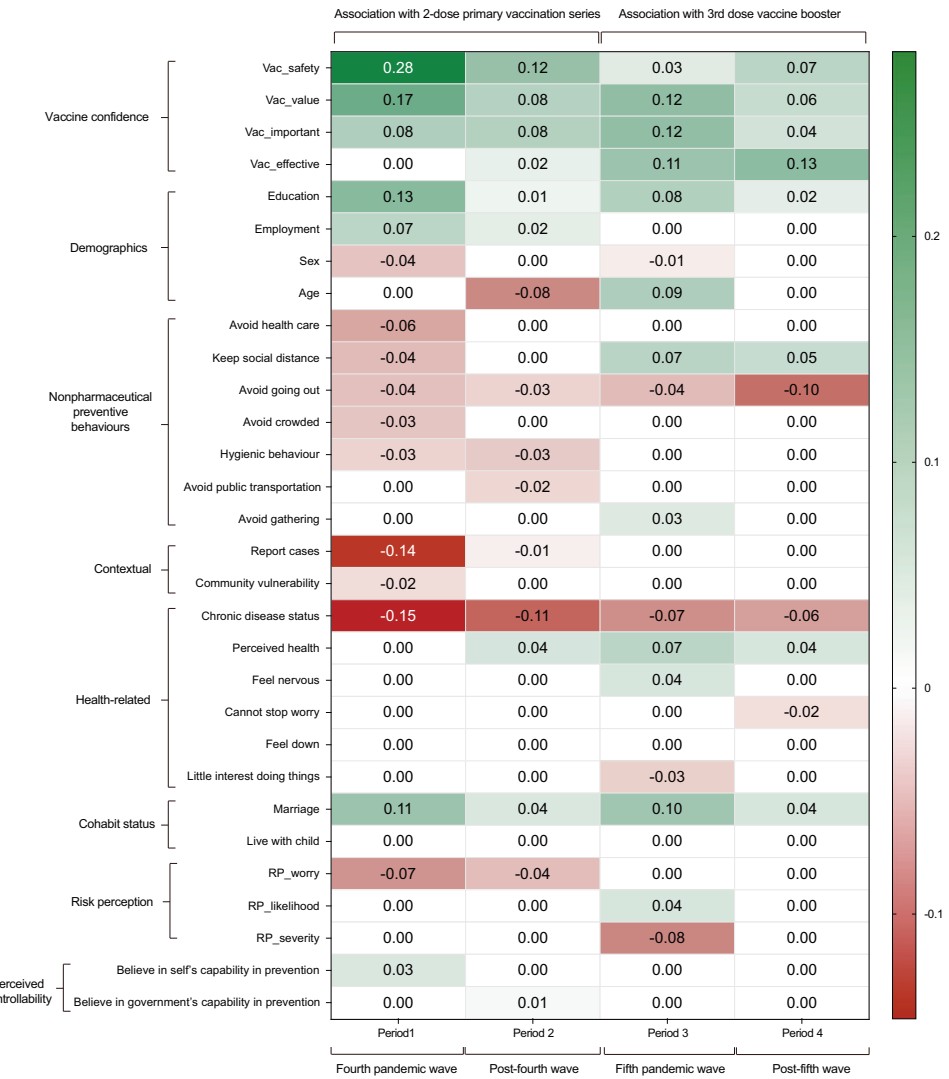

**Fig. 4 | Dynamic change of important determinants associated with vaccine uptake across P1-P4.** Numbers in the boxes are the absolute edge weights (standardised) over all variables' parameters across the four periods. Variables were ranked by the sum value of each domain in P1. Greater absolute number indicates a stronger association with the uptake of the primary dose series or the booster dose of COVID-19 vaccines, while zero indicates no direct association. The signs of edge parameters were obtained from the interaction function in mgm, green colour denotes positive association and red denotes negative association with vaccination uptake. We used abbreviations for some variables in the figure to ease interpretation. The four variables in the Vaccine Confidence domain are: "Vac_safety": COVID-19 vaccine is safe; "Vac_value": COVID-19 vaccine is compatible with personal value; "Vac_important": Taking COVID-19 vaccine is important; "Vac_effective": COVID-19 vaccine is effective. The three variables in the Risk Perception domain are: "RP_worry": Perceived worry of being infected with COVID-19; "RP_likelihood": Perceived likelihood of being infected with COVID-19; "RP_severity": Perceived severity of being infected with COVID-19.

from P1 to P4. Older age was found to be negatively associated with vaccine uptake in P2, but was positively associated with vaccine uptake in P3, when the most severe pandemic outbreak exploded in Hong Kong and caused more than 7300 deaths in unvaccinated older adults[47]. In response to the high COVID-19 risk, Hong Kong government launched the Home Vaccination Service to provide free door-to-door vaccination service to older adults aged 70 or above, pushing the at-risk group to take the booster vaccine dose[44]. Higher educational attainment and being married were associated with greater uptake of the primary vaccine doses and a booster dose. Such associations were particularly strong at the initial promotion of the uptake of the primary doses and the booster dose (P1 and P3) than in the scale-up phase (P2 and P4). This indicates that better-educated and married groups are more likely to be the innovation adopters of new vaccination policies or recommendations.

Number of COVID-19 cases and non-pharmaceutical preventive behaviours including avoidance behaviours and hygienic behaviours

had negative associations with vaccine uptake. This pattern was especially evident in P1 when most non-pharmaceutical preventive behaviours showed a negative association with the primary doses of vaccine uptake. P1 covers the fourth wave of pandemic when the cases number surged in the community, and it also covers the initial rollout phase of the vaccination programme. Several reasons may explain the negative association between adoption of non-pharmaceutical preventive behaviours and uptake of the vaccines in P1. First, since the vaccines were relatively novel for people in this period, people tended to overestimate the risk of the vaccines (i.e., vaccine safety and side effects) and hence chose to stick with status quo, which was the adoption of the safer and more familiar non-pharmaceutical self-protective measures such as social distancing measures. Second, as people had gotten used to non-pharmaceutical measures for self-protection over 1 year since the pandemic emerged, which together with the zero-COVID policy in Hong Kong may downplay people's perceived risk of the disease and the importance of vaccines. For some,

visiting a vaccination centre for taking a vaccine was perceived to be even more risky[14].

Furthermore, we found that "Trust in government" though showed no direct link with vaccine uptake, it is closely linked to vaccine confidence attitudes. Specifically, greater trust in government was associated with greater vaccine confidence, which in turn was associated with vaccine uptake (Fig. 3). "Trust in government" is also closely linked to various demographic variables, suggesting that it is an important bridging node that confers the effects of other nodes onto vaccine uptake.

## Verbal reasons for vaccine hesitancy by age groups

At the later phase of the vaccination programme when the vaccine was rolled out for 11 months, a total of 1043 participants who reported being hesitant or resistant for taking the vaccine primary doses or booster doses were asked about their hesitant reasons. Amongst them, 983 provided at least one reason (Supplementary Table 2). For uptake of the primary vaccine doses, the most frequently mentioned reasons were "concern about vaccine safety and effectiveness", "no urgency to take a vaccine", and "having got sufficient antibodies" (due to natural infections or getting one dose of vaccine) amongst younger groups aged between 18 to 44 years. The youngest age group (18–24) also mentioned that "lack of trust in government" was their major reason (22.2%) for resisting the primary doses of vaccine. Resistant reasons differed in older groups aged 45 years or above, with "worry about chronic disease status" ranking at the top, especially amongst those aged 65 years or above of whom 43.6% mentioned this reason. In addition, participants aged 65 years or above frequently mentioned "lack of recommendation and support from doctors, family and friends" as one major barrier (11.9%) to taking the primary doses of vaccine. For reasons of refusing the booster vaccine dose, the top three reasons were similar amongst all the age groups, with concerns about vaccine safety being the top concern (average 30.8%), followed by feeling no urgency or low need to take the booster (average 28.6%), and concern about vaccine effectiveness (average 21.6%). Notably, worry about chronic disease status was no longer a major reason for booster vaccination resistance in older adults. Alternatively, older people tended to refuse a booster dose due to optimistically believing that they already had sufficient antibodies obtained from natural infection or the first two doses of vaccines (10.8%).

## Sensitivity check

We first checked the stability of our networks. The accuracy of edge weights of our sample estimates consistently fell within the boot-strapped confidence intervals across the four networks (Supplementary Fig. 1–4), indicating that the edge weights of our sample estimate were overall accurate. We then conducted a sensitivity check using a 7-day time window for COVID-19 report cases and death numbers, the results remained robust (Supplementary Fig. 5).

## Discussion

Pandemic is a situation of high uncertainty, constant changes, and high personal threat. In this study, we utilised real-time population-based data that covered the entire period of the COVID-19 vaccine rollout, to systematically investigate determinants associated with the primary vaccine doses and booster dose uptake. Using multi-level variables, our study offers insights into how the vaccination programme in a pandemic context should adapt to the changing situation and related psychological responses to optimize vaccination uptake.

Building on existing literature that vaccine confidence plays the key role in people's COVID-19 vaccination decision[7,13,48], our findings provide more nuanced insights into the changing importance of vaccine confidence attitudes for vaccination uptake with the evolution of the vaccination programme. We found that at the initial stage when COVID-19 vaccines became available to the public, vaccine safety was

the most important determinant of primary vaccine doses uptake. Concerns about vaccine safety were mainly driven by the novelty of the vaccine technology and its rapid development[49], and further intensified by the negative news regarding COVID-19 vaccines and people's conspiracy beliefs[28,50]. As uptake of the primary doses became high, people gained more confidence about the safety of the vaccine. However, the resurgence of COVID-19 outbreaks due to the emergence of new virus variants and waning immunity[51] could dampen people's confidence in the vaccine effectiveness. In other words, people's attention shifted from the potential harms of the vaccine to its expected benefits in the later stage of the vaccination programme.

Our network analysis also consistently identified that persons with chronic disease status were less likely to take the vaccinations across the whole vaccination programme. Relatedly, older age was found to be negatively associated with vaccine uptake in P2, but the association shifted to be positive in P3. Participants' verbal reasons for vaccine hesitancy revealed that concern about chronic disease status was the main reason for refusing the primary vaccine doses uptake in older adults (Supplementary Table 2). This explains why uptake of COVID-19 vaccine was low among older people before P3 (the Omicron wave), with around 35%, 50%, and 80% of people aged 60–69, 70–79, and 80 years or above, respectively, had not received any dose of COVID-19 vaccines[44], resulting in high daily COVID-19 mortality[47]. The low vaccine uptake among older adults could be attributed to the special contexts in Hong Kong[17,52]. First, older people in Hong Kong particularly those with chronic diseases received insufficient information and no explicit advice from healthcare workers and their family members on COVID-19 vaccination[14]. This may link to the facts of lacking continuity in health care shaped by the existing healthcare system[53] and that most older people lived alone or with their older partners in Hong Kong[54]. Second, Hong Kong Chinese older adults generally have lower educational attainment, which limits their ability to utilise health information for making a medical decision. There were only 11% of Hong Kong older adults obtained the tertiary education in 2021[55], compared to 47.1%, 32.1%, and 38.8%, respectively, in Japan, New Zealand and the United Kingdom[56] where a higher willingness to take a COVID-19 vaccine was reported among older adults[19,57,58]. However, older age was positively associated with uptake of COVID-19 vaccine during P3, which may be attributable to the joint effect of intensive media attention on deaths among older people due to COVID-19 and the implementation of door-to-door mobilisation of vaccination for older people during this period[44].

Previous studies have shown that merely reporting daily case numbers has little impact on behavioural change, including vaccination[59,60]. Our study found that reporting the number of COVID-19 cases had a negative association with the primary vaccine doses uptake in P1 and P2, and the association with uptake of the booster dose disappeared in P3 and P4. The pattern at the very beginning could be linked to people's avoidance of public places (such as vaccination centres) to protect themselves against infection. However, the number of reported cases became insignificant in the later stage. This may be attributed to the growing resilience and familiarity of the prolonged public health crisis[61] and the fact that vaccination no longer aimed at preventing infection[62]. Contrary to previous observational studies concluding that more adoption of non-pharmaceutical preventive behaviours was associated with higher vaccine uptake[50,63], we observed a consistent negative association between the adoption of alternative protective behaviour (i.e., "avoiding going out") and vaccination uptake. In addition, adoption of an array of non-pharmaceutical preventive behaviours was all negatively associated with vaccination uptake at P1, when the vaccine was initially rolled out. Several reasons may explain this. First, people who avoided going out may perceive a higher risk of infection by visiting the vaccination venues. Second, people who avoided/were able to avoid going out may perceive a low risk of exposure to the viruses and hence perceived a

low need for vaccination. Third, the negative associations between different non-pharmaceutical preventive behaviours and vaccine uptake at P1 indicate that people tended to adopt alternative behaviours that they may perceive to be safer when evaluating the pandemic risk against the vaccine risk. Fourth, the booster dose was promoted through the implementation of vaccine pass which required people to take a booster dose for proof to access certain premises. This may induce psychological reactance, particularly among people with higher vaccine hesitancy[64]. Although there may be reverse causality that people who had received the vaccine tended to be more relaxed about taking the non-pharmaceutical measures, we ruled out this possibility by running additional chi-square tests between vaccine uptake status and adoption of non-pharmaceutical preventive behaviours. We found that the negative association was mainly driven by the greater proportion of adopting non-pharmaceutical preventive behaviours in the vaccination non-uptake group (see Supplementary Fig. 6). Overall, this finding suggests that the non-pharmaceutical preventive measures remained important for people who would like to avoid any vaccine side effects or other adverse effects[14]. Studies consistently found that people's positive traditional Chinese medicine (TCM) value can induce negative attitudes toward western biomedicine including vaccination[14,52,65]. In Hong Kong, some individuals, particularly older adults[14], are more familiar with TCM and perceive that TCM is less invasive compared with western biomedicine[66]. Adoption of non-pharmaceutical measures is important for pandemic control at the initial stage when vaccines are not available[12]. However, it also induces complacency psychology and illusory optimism that vaccines are no longer needed[67], which was found to be the main barrier for booster dose uptake in our qualitative analysis of participants' verbal reasons for vaccine hesitancy. In the later stage of a pandemic, strategies should focus on mitigation rather than containment, during which stringent social distancing measures may induce tremendous societal costs and thereby vaccination is of paramount importance particularly for individuals at higher risk of severe disease[47]. In the later stage, an illusory belief about the effectiveness of non-pharmaceutical preventive behaviours may be detrimental to promoting vaccination uptake[68].

We also identified potential innovation adopters that could be targeted at the early stage of a vaccination campaign. Our results suggest that people with higher educational attainment and reported married status were more likely to take the primary vaccine doses in P1 and a booster dose in P3 when the new recommendation or policy for vaccination was introduced. People with higher education have a greater ability to comprehend new interventions and policies and are thereby usually the early innovation adopters[69,70]. While married people's early adoption of novel intervention is likely to be driven by prosocial motivation, to take the risk and endure the uncertainty of vaccine safety to protect their loved ones[17]. Prior studies found no clear associations between educational attainment and marital status with COVID-19 vaccination acceptance in their survey time[7,20]. Our study found that these demographics only predict vaccination uptake at the initially established phase of the vaccination campaign when taking the vaccination was yet to be normative. Future programmes can leverage these early adopters to spread the pro-vaccination norm in the whole population[71].

Network results revealed that trust in government is an important bridging node that connected individual and interpersonal factors with vaccine-specific factors of confidence attitudes which in turn are directly and strongly associated with vaccination uptake. Specifically, individual demographics and perceived risk of COVID-19 and self-efficacy in preventing COVID-19 were first linked to trust in government which further connected to vaccine confidence attitudes. This indicates that the effects of individual factors on COVID-19 vaccine hesitancy and uptake could be partly because trust in government varied by these individual factors. The analysis of the verbal reasons of

vaccine hesitancy suggested that distrust in government was a frequently mentioned reason for refusing the primary vaccine doses uptake among the youngest people (aged 18–24). We also found that distrust in the government tended to be co-mentioned with other reasons. Both the network analysis and the analysis of the verbal reasons consistently indicate that trust in government is important to bridge different determinants of vaccination uptake, highlighting the importance of building trust for addressing vaccine hesitancy and improving vaccination uptake across individuals of different characteristics.

This study has both theoretical contributions and practical implications for informing more efficient vaccination programme in the future. One recent systematic review including 47 studies concluded that there are multiple determinants underlying COVID-19 vaccine hesitancy involving individual vaccine confidence beliefs, trust in authorities, self-efficacy, information influence, emotional state (i.e., fear and anxiety), and social influence[72]. A strength of the current study is that it considered the complexity of vaccination decisions through a network lens, which allowed us to depict the complex interactions among multiple determinants associated with vaccine uptake. The included determinants can be mapped onto multilevel: individual/interpersonal level (i.e., COVID-19 vaccine confidence, trust in government, demographics), contextual level (i.e., residential community vulnerability level, pandemic-related situation evolution such as numbers of COVID-19 cases and deaths), and vaccine-specific level (i.e., COVID-19 vaccine confidence attitudes). The various data sources enable us to construct relatively comprehensive models to understand vaccination uptake. Another strength of this study is the dynamic perspective in investigations of COVID-19 vaccination. This is especially relevant as pandemic circumstances involved constant changes of multiple contexts including disease incidence in the community, media focus, policies, control measures and associated public risk perceptions. The dynamic view and investigations warrant a more accurate picture of real-time public concerns and agenda on COVID-19 and its vaccines, thereby providing timely insights and instructions for effective risk communication and vaccination promotion.

Our study has several limitations. One limitation of this study is that in the later stage of the pandemic when the society gradually returned to new normalcy, we did not measure the full set of variables at bi-weekly basis in P4, which resulted in only 2 weeks' data included in this period. However, the sample size in each survey week remained sufficient to estimate population characteristics (see Methods). Second, vaccination uptake was self-reported. Despite this, we found a high correlation between the self-reported vaccine uptake rates in our surveys and the actual vaccine uptake rates reported by the government ($r(15) = 0.995$, $p < 0.01$; see Supplementary Table 7), indicating that the self-reported vaccine uptake was a reliable indicator of actual vaccination uptake. Third, the repeated cross-sectional study design does not allow us to establish causal relationships of determinants with vaccine uptake. However, our study design answers the relative importance of determinants and is suitable for surveillance on vaccine uptake among the public throughout the vaccination campaign period. It supplemented the shortcomings of high attrition rates and costly efforts in maintaining the cohort across consecutive time points. Fourth, we only measured people's COVID-19 vaccine confidence attitudes in our survey, therefore, conclusions of current study might not be able to be generalised to other vaccination contexts. Fifth, we excluded participants with linguistic and cognitive problems because these participants may not provide clear responses over a telephone interview. Such exclusion limits our understanding of vaccine hesitancy in these minority groups. Last, our networks did not exhaustively include all variables that were associated with vaccination uptake which contributed to the relatively low predictability of the networks for vaccine uptake. Similar studies have reported explained variances ranging from 10% to 78%[73–76]. However, those studies used vaccine

intention instead of vaccine uptake as the main outcome. Comparatively, our study showed a similar explained variance of ~13% as another study that reported 15% of explained variance in vaccine uptake[77].

Hong Kong's experience in meeting the challenges of the COVID-19 pandemic has implications for future vaccination campaign against a pandemic in other regions of the world except when contexts are highly divergent. First, communications should highlight and address the salient attributes of the vaccine concerns dynamically. While communicating about vaccine safety is important at the early stage of the vaccination campaign, reinforcing the public's confidence in vaccine effectiveness should be prioritised for promotion of booster dose uptake. This can be done by giving timely feedback on how vaccine uptake helps to reduce people's risk of infection or disease complications. Second, there could be complacency psychology and illusory optimism due to overconfidence in the effectiveness of non-pharmaceutical preventive behaviours, which could dampen motivation for vaccination when the vaccines become available. This is particularly the case when the disease incidence in the community is kept at a low level with the implementation of stringent social distancing measures. Vaccination campaigns should highlight the unique contribution of vaccine uptake, the potential societal costs of prolonged social distancing measures, and the importance of mitigation rather than containment at the later stage of the pandemic. Third, it is possible to leverage the early innovation adopters including those with higher educational attainment and who are married to make their vaccination decision more visible and positive to other wait-and-see groups. Fourth, it is important to establish trust in the public to promote vaccine uptake by enhancing people's confidence in the COVID-19 vaccines. A possible approach to establish trust is through partnerships with influential figures such as political figures of various political ideologies to reduce hesitancy and mitigate the polarisation of vaccines[78]. Fifth, older people were identified as the most hesitant group to take a novel vaccine, possibly attributing to the contexts in Hong Kong. Interventions should specifically focus on older adults and persons with chronic conditions to reduce their vaccination concerns. A potentially effective approach is to leverage the doctors' long-standing relationship with older patients to clarify the safety of COVID-19 vaccines and address their concerns about their weak physical function to endure the vaccine side effects[14,17]. Overall, future vaccination campaigns should timely identify and respond to the various determinants of vaccine uptake by periodically reviewing the evolution of the pandemic to mitigate potential loss brought by people's unnecessary delayed vaccination decisions.

## Methods

### Survey data

Data were extracted from populational-based repeated cross-sectional surveys conducted on a weekly or monthly basis to monitor acceptance of COVID-19 vaccine among the general adults since the COVID-19 vaccination campaign was launched in early 2021 in Hong Kong[7,12,79]. In each survey round, we recruited Hong Kong adults aged 18 years or above using random-digital-dialled telephone interviews with a ratio of 1:1 for landlines and mobile phones. Telephone interviews were conducted using Cantonese or Mandarin which covered over 92% of the population in Hong Kong[55]. Individuals with linguistic and cognitive difficulties to complete a telephone interview were excluded. Verbal informed consent was collected from eligible individuals before data collection. The target sample size was an alternative of 500 or 1000 on a regular interval, which was sufficient to estimate population characteristics ($p = 0.5$) with a margin of error of 0.04 and 0.03, respectively, and a 95% confidence interval ($t = 1.96$)[7]. Details of sample size and the response rate for each survey round can be found in Supplementary Table 3. In each round, core study measures such as uptake of COVID-19 vaccines and non-pharmaceutical preventive measures were retained throughout while study measures such as COVID-19 vaccine confidence variables were rotated to maintain a feasible length of the questionnaire for a telephone interview. We included survey rounds that contained the core and identical study measures to optimize the comparability of networks across different periods. Totally, data from 17 survey rounds (each round of data collection lasted for 3–4 days) were used for the current network analysis including four rounds for P1, seven rounds for P2, four rounds for P3, and two rounds for P4. Survey timeline covered an initial period of vaccine rollout (Mar 2021), a scale-up period for promoting the completion of the two primary doses of vaccines, an initial phase of recommending the booster vaccination, and a scale-up period for promoting the completion of the booster dose (November 2022). This study received ethical approval from the Institutional Review Board of the University of Hong Kong/Hospital Authority Hong Kong West Cluster (Reference No.: UW 20-095).

### Contextual data

To map relevant factors on the contextual level, we also retrieved data from multiple sources to measure the external environmental influences on people's vaccine uptake. Data on daily COVID-19 reported number of cases and deaths were obtained from the Hong Kong Centre for Health Protection[80]. Community-level data were used to construct the social vulnerability index (SVI) following our previous study[81] using data obtained from the 2021 Hong Kong population census data[55]. In case where the most recent census data were yet to be publicly available, the 2016 by-census data[82] were used instead to construct the SVI. The dataset used for SVI construction is provided in Supplementary Table 4.

### Study instrument

Details of our study instrument and coding strategy are provided in Supplementary Table 5. To briefly summarise, a multitude of determinants that could potentially explain COVID-19 vaccine uptake were included in the network analysis, which were determined based on several systematic reviews on determinants of COVID-19 vaccine hesitancy[29–31] and mapped onto the framework of vaccine hesitancy determinants proposed by the WHO SAGE Working Group[11]. The framework suggested that vaccine hesitancy is multifaced, shaped by multi-level determinants, ranging from individual/interpersonal factors to vaccine/vaccination specific and contextual factors[21]. To capture the complex interactions amongst multilevel factors in shaping vaccine uptake, we used a blending of survey-based data and real-world time-varying contextual data to construct our network models. The surveys collected individual, interpersonal, and vaccine-specific factors, while data on contextual factors were obtained from publicly available official data sources.

**Vaccine specific factors.** Although vaccine accessibility plays an important role in determining vaccination behaviour, it was not applicable in Hong Kong because Hong Kong had procured sufficient vaccine doses for all its residents and made it easily accessible through setting multiple community vaccine centres. Therefore, we collected vaccine confidence variables (perceptions of vaccine safety, efficacy, importance and value alignment)[48] as indicators of vaccine-sepcific factors.

**Individual factors.** This included: (1) risk perception of COVID-19 (perceived personal vulnerability, severity of the diseases and worry about infection)[7,12]; (2) perceived controllability of pandemic control including trust in government in controlling the pandemic and perceived self-efficacy in preventing the infection)[7,12]; (3) physical health status including chronic disease status and self-reported health status[14,17,18]; (4) psychological distress (measured with PHQ-4)[19,20]; (5) non-pharmaceutical preventive behaviours including hygienic practice and avoidance behaviours[13,14]; (6) demographic variables: age, gender, education, and employment[7,10,20].

**Interpersonal factors.** Given that vaccination is not only an individual decision but also a prosocial behaviour to protect important others[83], we also included two cohabit characteristics to represent the interpersonal factors: (7) marriage status[17] and (8) whether living with children[22].

**Contextual factors.** we included the pandemic situation (official reports on number of cases and deaths)[25] and the vulnerability of people's residential community[20,26,27]. The data relating to the contextual factors were retrieved from official sources for three contextual determinants: (1) average number of COVID-19 report cases 2 weeks before the survey date; (2) average number of deaths caused by COVID-19 2 weeks before the survey date; and (3) people's living community's vulnerability. We included one's living community as one important determinant because studies consistently found that people who live in more vulnerable neighbourhood had lower vaccination uptake rates[20,26,27] and the communities they lived in were more likely to form local outbreaks. To construct this determinant, we used the participant's self-reported living district to construct a SVI to indicate the vulnerability of each participant's living community. Details of SVI construction methods had been reported elsewhere[20,81], the ranking details can be found in Supplementary Table 6. We did not include policy-related determinants in our network models because participants in each pandemic period could experience the same policies, thus, the impact will be homogeneous. Furthermore, government policies typically fall behind the actual situation, and it takes time for the public to receive and decode the policy information and act accordingly.

**Supplementary data.** As a supplement to interpret the network findings, we also collected verbal responses from participants to understand their specific vaccine hesitant reasons at a later phase of the vaccination programme. In nine survey rounds conducted between 6 December 2021 and 14 July 2022 (11 months after the vaccine was rolled out), participants who had not received any/received only one dose/received only two doses of COVID-19 vaccines but indicated that they would be never/very unlikely/unlikely/unsure to take one dose/ the second dose/the booster dose of COVID-19 vaccine in the future were further asked about the major reasons for being hesitant or resistant about taking COVID-19 vaccination. The open-ended responses for why to be hesitant or resistant about taking a COVID-19 vaccine or a vaccine booster from these nine survey rounds were coded as reasons for vaccination resistance. Participants were asked to provide reasons that first came to mind, then the interviewer jot down notes of participants' statements and asked follow-up questions of "any other reasons" to encourage participants to give more than one reason for their vaccination decision.

**Outcome measurement**

Different outcomes were used to represent the dynamic change of a vaccination campaign. The COVID-19 vaccination programme in Hong Kong initially targeted promoting the completion of the primary series of vaccine doses from 26 February 2021. On 3 November 2021, 10 months after the programme had been launched and the completion of the first dose and second dose had reached above 68% and 65%, respectively[44], the Hong Kong government started to focus on the promotion of the vaccine booster uptake initially in high-risk groups including immunocompromised patients, persons with chronic conditions, and older adults aged 60 or above, and 3 weeks later expanded to general adults who had completed their primary doses uptake for at least six months[44]. Following this timeline, we used the booster dose uptake as the outcome of vaccination uptake if the participant completed the survey on or after 3 November 2021 (covered P3 and P4). While participant's primary vaccine doses uptake was used as the outcome of vaccination uptake if they completed the study before 3

November 2021 (covered P1 and P2). Primary vaccine doses uptake was indicated by the first-dose vaccine uptake suggested by other studies[17,84]. Booster dose uptake was defined as the third dose of vaccine uptake after completion of the primary series[17,41]. All vaccination uptake outcomes were self-reported, which differed from other studies that combined high vaccination intention with actual uptake behaviour[6,19,48]. A recent study suggested that intention cannot always reflect actual COVID-19 vaccination behaviour[85]. Our analyses indicated that the self-reported vaccine uptake rates based on our survey were highly consistent with the actual vaccine uptake rates amongst eligible population reported by the Hong Kong government, suggesting that self-reported vaccination uptake was a good proxy for actual vaccination behaviour (see Supplementary Table 7).

**Statistical methods**

**Descriptive analyses and data pre-processing.** We first provided distributions of participants' sex, age educational attainment, and employment status across the four study periods (P1-P4) and compared that with the most recent census data[55]. Proportions and the corresponding 95% confidence intervals of each dose of vaccination uptake across P1-P4 were weighted to the population's age and sex using the 2021 census data, respectively. For contextual data—the number of COVID-19 cases and death numbers in the community, were obtained from the official source and plotted the number of report cases from 23 Jan 2020 to 31 Dec 2022. To account for the potential impact of contextual determinants on COVID-19 vaccine uptake, we used a 14-day time window in our main analyses. We also conducted sensitivity checks on 7-day time window, the results remained robust (Supplementary Fig. 5). The number of COVID-19 report cases and deaths were averaged over a 14-day period. Besides, before running the network analysis, we checked the proportion of missing values in all the selected variables. We removed two variables in the network models that had a high proportion of missing values (monthly household income, missing rate 20.8%; one vaccine confidence item, missing rate 28.7%; Supplementary Fig. 7). Then multiple imputation was conducted in R using mice package[86] to replace a small proportion (no more than 8%) of missing values in other variables, using predictive mean matching approach for five iterations with 20 imputations.

**Network graphs.** Data analysis was carried in R version 4.2.3. All models were visualised as network graphs, with 'nodes' representing variables and 'edges' representing the conditional dependency between the variables[33,87]. The edge width was intuitively interpreted as the strength magnitude between the variables. The network layout was based on the algorithm of Fruchterman and Reingold from qgraph R-package[88], it generates plots with the most strongly associated nodes being placed at the centre of the graph and weakly associated ones at the periphery.

**Network estimates.** A mixed graphical model (MGM) was applied given that our model involved variables of categorical, continuous and count data[33]. One advantage of the MGM model is that it does not require an a priori commitment to any particular data-generating mechanism[89]. MGM can handle various types of data without unnecessary data transformation. In addition, unlike other models that focus on one specific variable or outcome at a time, MGMs can explore the relationships between all variables simultaneously[33,90]. In MGM, edges are parameterised as regression coefficients from generalised linear regression models. We used R-package *mgm* to estimate the pairwise weighted adjacency matrix amongst variables[33], then qgraph package was applied to visualise such edge-weights matrix as a network[88]. We adopted the penalty approach, namely, the *least absolute shrinkage and selection operator* (LASSO), to obtain a more conservative network estimation[91]. The LASSO approach shrinks edge weights by setting smaller edges to zero, thereby reducing the chance of getting false-

positive findings. For current study, we used the Extended Bayesian Information Criterion and set its hyperparameter to 0.50 to obtain a succinct network[92]. Codes for replicating all the results are available on the Open Science Framework: https://osf.io/r58e7/.

**Relative important determinants of vaccine uptake.** As mentioned above, the edge weights can be interpreted as the strength of conditional dependency. Therefore, by assessing the edge weights associated with the vaccine uptake node we can provide intuitive interpretations of the relative importance of various determinants on vaccine uptake in the networks. We ranked the absolute edge weights of all variables that had an edge with the node of vaccine uptake across P1–P4 to assess the dynamic change of the relative importance of determinants associated with vaccine uptake. We extracted the sign (direction of the association) and the relative importance of continuous variables from the weighted adjacency matrix. However, for categorical variables with more than one level (i.e., one parameter for each of the two or more categorical levels), the weight of the edge for categorical variables was measured by the mean of the absolute value of all parameters. Because the signs for categorical variables cannot be directly shown in the graph (they were manifested as grey lines), we further extracted the specific interaction and their direction from the parameterised multinomial regression in *glmnet*, which models the probability of each level of the categorical predicted variable with the first category of the predictor variable as the reference category[33]. The reference category for categorical variables can be found in Supplementary Table 5.

**Predictability estimates.** Predictability in network analysis refers to how well a node is predicated by all other nodes in the network[93]. The predictability for categorical variables was estimated using the normalised accuracy, this indicator was computed by the probability of the node that can be predicted by all other nodes in the network after removing the influence that is achieved by the trivial prediction, while that for continuous variables was indicated by the proportion of explained variance ($R^2$). Results were visualised using a grey pie chart surrounding the nodes in the network, with a larger shaded area indicating greater predictability.

**Stability.** Post-hoc stability analyses for edge-weight parameters were conducted to assess the reliability of the network estimation. The R-package *bootnet* was used for the stability check[91]. The accuracy of edge weight estimates was inferred by calculating the 95% confidence interval (95% CI) of the weight estimates using the non-parametric bootstrap. Wider 95% CI indicates more unstable estimate.

**Coding of the verbal reasons for vaccine hesitancy.** To reduce redundancy, similar reasons for being hesitant or resistant to take a COVID-19 vaccine were combined to represent one reason category. For instance, "I am afraid of vaccine quality" and "I worry about vaccine safety" were coded as "Concerned about vaccine safety". Two coders (J.Y. and Y.X.) coded the reasons independently and any disagreements were resolved before finalising the coding scheme. Percentage of each reason category was calculated by age groups and by the primary doses and booster doses of vaccine uptake.

## Data availability
The data of our study are publicly available in an OSF repository: https://osf.io/r58e7/. There are no restrictions to accessing the data. The de-identified verbal reasons for individual vaccine hesitancy can also be accessed through OSF. Data for Social Vulnerability Index construction were retrieved from several sources (https://www.census2021.gov.hk/en/district_profiles.html AND https://www.censtatd.gov.hk/hkstat/sub/so459.jsp AND https://www3.ha.org.hk/data/HAStatistics/StatisticalReport/2020-2021). Data of COVID-19

report cases and death numbers were retrieved from official source (https://data.gov.hk/en-data/dataset/hk-dh-chpsebcddr-novel-infectious-agent (2023)).

## Code availability
The analysis codes are publicly available in the same OSF repository: https://osf.io/r58e7/.

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

## Acknowledgements

This study received funding from the Health and Medical Research Fund, the Health Bureau, The Government of the Hong Kong Special Administrative Region (Ref. No.: COVID19F04; COVID19F11).

## Author contributions

Q.L. and B.J.C. designed and oversaw the study. J.Y. and Y.X. contributed equally to data collection, data analysis and data interpretation under the supervision of Q.L. J.Y., Y.X. and Q.L. wrote the manuscript. I.O.L.W. contributed to participant recruitment, project management and revising the manuscript. W.W.T.L. and M.Y.N. contributed to the interpretation of the findings and revising the manuscript. All authors approved the submitted version and have contributed to the final version of this manuscript.

## Competing interests

B.J.C. consults for AstraZeneca, Fosun Pharma, GSK, Moderna, Pfizer, Roche, and Sanofi Pasteur. The authors report no other potential competing interests.
