## [Peer Review File · Nature Communications]

Dynamic predictors of COVID-19 vaccination uptake and their interconnections over two years in Hong KongREVIEWER COMMENTS

Reviewer #1 (Remarks to the Author):

This study looked at predictors of COVID-19 vaccine uptake in Hong Kong, and how these predictors changed over the course of the pandemic.

I have some comments the authors may wish to consider when revising their manuscript. I have structured my comments by paper section.

Introduction

I found the Introduction a bit hard to follow. I would provide a more concise overview of known determinants of vaccine confidence and uptake (there are notable studies / literature reviews from the WHO, Heidi Larson and team, and others). The authors identify some relevant factors, but I would present this information in a more systematic and clear manner, since this is very important for understanding the study methods and choice of variables. There are several frameworks that nicely group factors into key categories (as explained in the methods under the "Study instrument" section – I think this can be moved to the introduction). Otherwise, it's unclear why the authors immediately discuss the importance of communication before later saying it's essential to understand the influences of vaccine confidence to guide vaccination programs.

After providing a more general / generic overview, the authors can identify a few key issues that were relevant with COVID-19 specifically (and Hong Kong specifically). As I said, some of this was covered in the current intro, but I would urge the authors to think carefully about how they can present this in a more logical and concise manner.

I would also summarize existing literature in the introduction. Other academics have studied drivers of covid-19 vaccine confidence (in both quantitative and qualitative studies). I would highlight these studies to identify a research gap that the authors seek to fill.

Methods

I am not a statistician so I would urge the editors to carefully consider the feedback from other reviewers regarding the methods. While the methods seem appropriate, I think it is important that a statistician review the sampling approach, network analysis, imputation technique, etc., to ensure the results are valid.

The authors may want to think about a way to more objectively justify the choice of indicators in their instrument. The authors explain that they conducted a literature review (citing a number of studies) and then outline the 10 groups of indicators they considered. Because the authors don't outline any systematic approach to constructing the instrument, I would imagine that another team might arrive at different set of variables. Perhaps the authors can outline the main framework they used to construct this instrument, and clearly outline any deviations / additional factors considered. I don't conduct this type of research myself, so I don't exactly know what the norm is for this type of research (and the editors can tell the authors if they should disregard my comment), but it feels like a somewhat subjective approach to constructing the instrument (even if most of the variables seem sensible).

Did the model reflect the availability of vaccines in the Hong Kong, which would probably explain uptake amongst some individuals?

In the methods, I think it would help to have topic sentences to make clear why different data were collected. Otherwise, it's not clear in the methods why mortality data was needed if the authors were interested in looking at determinants of vaccine uptake (before getting to the instrument section, where the authors repeat that they collected case / mortality data).

Results / Discussion

I don't have any major comments about these sections, which followed naturally from the methods. Having said that, I think the results would be even easier to follow if the instrument is justified more systematically (and relevant background info is contained in the intro). Then it will be easier for the reader to understand why certain associations were observed. As it stands, I couldn't immediately understand why some factors seemed to drive vaccine uptake (although the authors later did a good job of contextualizing their findings in the Discussion and outlining policy implications).

Reviewer #2 (Remarks to the Author):

The study sampled a representative, large sample of Hong Kong residents in several waves across major stages of the COVID-19 pandemic to systematically study the dynamic determinants and their interactions associated with the COVID-19 primary vaccine and booster uptake at different stages of the COVID-19 vaccination campaign. It used a mixed graphic network model to depict the association between vaccine uptake and a comprehensive set of individual, community, and more extensive social contextual variables and their complex interactions. It also cooperates a qualitative part to describe participants' reasons for COVID-19 vaccine hesitancy. They found the shift of public belief in vaccine safety at the early stage of a pandemic to the vaccine effectiveness at a later stage associated with vaccine uptake. In addition, they found that higher education and married status are associated with the early adoption of new vaccines, which can be an implication and target group for future new vaccine introduction.

In the Introduction or methods, could you provide more information on the rationale for why you used this particular model?

In the methods (data collection), you describe your measurements of vaccine hesitancy. The study targets the COVID-19 vaccine specifically, and the COVID-19 vaccine could have a different set of attitudes and hesitations as a result of its quick rollout and its unprecedented promotion campaign. As we understand it, the study did not collect any variables that related to regular vaccine hesitancy. Thus, we cannot differentiate the COVID-19 vaccine hesitant participants were hesitant to all vaccines or only COVID-19 vaccine specifically. Does that affect your interpretation of your results?

Methods- Line 471: Excluding participants with linguistic and cognitive problems could lead to bias since these populations usually delay vaccination and have lower vaccination rates. Limitations of excluding these people need to be discussed. Additionally, was the telephone survey done in Cantonese or Mandarin or both?

Results- is it possible to provide a table showing vaccination coverage by different individual-level (i.e., demographic variables, trust in governments, etc.) and community-level attributes at different stages? This could let readers intuitively and statistically see changes.

Results-Line 248-249. The network results show that "trust in government" is not directly associated with the outcome of vaccine uptake but through vaccine confidence. However, in the qualitative part, "trust in government" is among the main reasons for COVID-19 vaccine hesitancy. Do you see this as an inconsistency?

Discussion- Line 364. Could you be more specific about vaccine risk? Were people worried about the side effects, whether the vaccine is not safe, or the vaccine is not effective enough to prevent them from COVID-19 infection?

Discussion- There is not enough discussion of the interaction of determinants, which is part of the aim of the study.

Reviewer #3 (Remarks to the Author):

The manuscript focused the problem of identifying key determinants and their interactions associated with the primary and booster vaccination uptake at different stages of the COVID-19 vaccination campaign. The authors considered a set of variables of different types, including categorical, continuous and count data. The authors employed an existing R-package on mixed graphical model estimation, and an existing R-package using non-parametric bootstrap to provide confidence intervals of the edge weights for the estimated networks. The authors also provided their code for the network estimation and analysis. The analysis looks okay to me.

Point-by-Point response

Reviewer 1:

This study looked at predictors of COVID-19 vaccine uptake in Hong Kong, and how these predictors changed over the course of the pandemic.

I have some comments the authors may wish to consider when revising their manuscript. I have structured my comments by paper section.

Introduction

I found the Introduction a bit hard to follow. I would provide a more concise overview of known determinants of vaccine confidence and uptake (there are notable studies / literature reviews from the WHO, Heidi Larson and team, and others). The authors identify some relevant factors, but I would present this information in a more systematic and clear manner, since this is very important for understanding the study methods and choice of variables. There are several frameworks that nicely group factors into key categories (as explained in the methods under the “Study instrument” section – I think this can be moved to the introduction). Otherwise, it’s unclear why the authors immediately discuss the importance of communication before later saying it’s essential to understand the influences of vaccine confidence to guide vaccination programs.

After providing a more general / generic overview, the authors can identify a few key issues that were relevant with COVID-19 specifically (and Hong Kong specifically). As I said, some of this was covered in the current intro, but I would urge the authors to think carefully about how they can present this in a more logical and concise manner.

I would also summarize existing literature in the introduction. Other academics have studied drivers of covid-19 vaccine confidence (in both quantitative and qualitative studies). I would highlight these studies to identify a research gap that the authors seek to fill.

Response 1. Thank you for taking time to review our manuscript and provide valuable comments. Based on the suggestion, we have restructured the introduction to follow a systematic approach to summarize current literature on determinants of COVID-19 vaccine hesitancy. Specifically, we use the framework proposed by the Strategic Advisory Group of Experts (SAGE) on Immunization Working Group¹ as a basic to summarize determinants of vaccine hesitancy or uptake belonging to individual, interpersonal, vaccine/vaccination-specific, and contextual factors identified in existing literature. This provides a rationale for the selection of determinants of COVID-19 vaccine uptake in our study. Based on the literature review, we highlight the limitations of separating contextual factors from individual/interpersonal and vaccine-specific factors in studying vaccine hesitancy, and explain the strength of using a network approach. Thereafter, we explain why vaccine hesitancy is dynamic through discussing evolving contexts in general and contextual factors specific to Hong Kong, based on which, we highlight why it is essential to monitor and track the dynamics of determinants of COVID-19 vaccine uptake. We have also revised our **Methods** section to make it consistent with the logic presented in the Introduction. To avoid repetition here, please find our revised arguments in the **Introduction and Methods** section, with red highlighting.

Methods

I am not a statistician so I would urge the editors to carefully consider the feedback from other reviewers regarding the methods. While the methods seem appropriate, I think it is important that a statistician review the sampling approach, network

analysis, imputation technique, etc., to ensure the results are valid.

The authors may want to think about a way to more objectively justify the choice of indicators in their instrument. The authors explain that they conducted a literature review (citing a number of studies) and then outline the 10 groups of indicators they considered. Because the authors don't outline any systematic approach to constructing the instrument, I would imagine that another team might arrive at different set of variables. Perhaps the authors can outline the main framework they used to construct this instrument, and clearly outline any deviations / additional factors considered. I don't conduct this type of research myself, so I don't exactly know what the norm is for this type of research (and the editors can tell the authors if they should disregard my comment), but it feels like a somewhat subjective approach to constructing the instrument (even if most of the variables seem sensible).

Response 2. Thank you for the valuable comment. As mentioned in Response 1, we have restructured our introduction and used the framework proposed by the Strategic Advisory Group of Experts (SAGE) on Immunization Working Group¹ as a basic to summarize determinants of vaccine hesitancy or uptake belonging to individual, interpersonal, vaccine/vaccination-specific, and contextual factors identified in existing literature. This provides a rational for the selection of determinants of COVID-19 vaccine uptake in our study. Based on the literature review, we highlight the limitations of separating contextual factors from individual/interpersonal and vaccine-specific factors in studying vaccine hesitancy, and explain the strength of using a network approach.

Did the model reflect the availability of vaccines in the Hong Kong, which would probably explain uptake amongst some individuals?

Response 3. Thank you for the comment. In the **Introduction**, we have clarified that “Sufficient vaccines for the whole population of over 7 millions were procured and made free and easily assessable for the population by setting up multiple vaccination centres in the community². Therefore, vaccine accessibility was not considered to be a major barrier to vaccination uptake in this study.” In addition, although the Hong Kong government gradually expanded the priority groups for COVID-19 vaccination, it took a relatively short time before the vaccination recommendation covered all eligible citizens. Hence, we believed that accessibility is not a major barrier. We have revised **Fig.1** to incorporate details about the timeline of changes in eligibility of COVID-19 vaccination from priority groups to the whole eligible population. Specifically, for the primary two-doses vaccine uptake, vaccinations were first arranged for five high-risk groups since the end of February 2021 including healthcare workers and people who participated in anti-epidemic works, older people aged 60 or above, people who worked or resided in a residential care home, people who worked for critical public services, and people whose work required cross-border commuting. The vaccination recommendation had been expanded to all adults aged 30 or above since 15 March 2021, and all residents aged 16 or above since 15 April 2021². For the booster dose vaccine uptake, the government first promoted a third-dose booster uptake in high-risk groups on 3 November 2021, which was expanded to all citizens who finished their primary doses uptake for at least six months on 23 November 2021². According to these timelines, both primary vaccine doses and booster dose were made available citywide within weeks of the rollout. However, we did believe that people who were physically incapable to travel to the nearby vaccination centre can be a barrier. Therefore, “chronic disease status” was included as one determinant in the network.

In the methods, I think it would help to have topic sentences to make clear why different data were collected. Otherwise, it's not clear in the methods why mortality data was needed if the authors were interested in looking at determinants of vaccine

uptake (before getting to the instrument section, where the authors repeat that they collected case / mortality data).

Response 4. Thank you for the suggestion. We have included subtitles for study measures at each level based on the framework proposed by the Strategic Advisory Group of Experts (SAGE) on Immunization Working Group¹ in the **Methods** section to improve the clarity.

Results / Discussion

I don't have any major comments about these sections, which followed naturally from the methods. Having said that, I think the results would be even easier to follow if the instrument is justified more systematically (and relevant background info is contained in the intro). Then it will be easier for the reader to understand why certain associations were observed. As it stands, I couldn't immediately understand why some factors seemed to drive vaccine uptake (although the authors later did a good job of contextualizing their findings in the Discussion and outlining policy implications).

Response 5. Thank you. We have restructured our **Introduction** to improve the clarity (See Response 1).

Thank you once again for providing all the valuable comments and advice for improving our manuscript.

Reviewer 2:

The study sampled a representative, large sample of Hong Kong residents in several waves across major stages of the COVID-19 pandemic to systematically study the dynamic determinants and their interactions associated with the COVID-19 primary vaccine and booster uptake at different stages of the COVID-19 vaccination campaign. It used a mixed graphic network model to depict the association between vaccine uptake and a comprehensive set of individual, community, and more extensive social contextual variables and their complex interactions. It also cooperates a qualitative part to describe participants' reasons for COVID-19 vaccine hesitancy. They found the shift of public belief in vaccine safety at the early stage of a pandemic to the vaccine effectiveness at a later stage associated with vaccine uptake. In addition, they found that higher education and married status are associated with the early adoption of new vaccines, which can be an implication and target group for future new vaccine introduction.

In the Introduction or methods, could you provide more information on the rationale for why you used this particular model?

Response 1. Thank you for taking time to review our manuscript and provide valuable comments. We have included more information on the rationale of using a mixed graphic network in the **Introduction**, with texts read: “Contextual determinants of vaccination uptake received increasing attention but were usually studied separately with other levels of determinants^{15,20,27}. It is suggested that individual and vaccine-specific factors are interconnected to co-shape people’s vaccination decision³². There remain limited understanding about how the contextual factors interact with individual and vaccine-specific factors. In this study, following the framework proposed by WHO SAGE on Immunization Working Group¹¹, we adopted a network approach to map individual, interpersonal, vaccine-specific and contextual determinants of vaccine hesitancy simultaneously³³. Compared with

conventional statistical models such as multivariable (linear or logistic) regression models, a mixed graphical network approach has the strengths of incorporating various data formats and allowing mutual interactions amongst variables and hence can reveal complex pathways lying between multilevel factors and vaccine uptake³³.”

In the methods (data collection), you describe your measurements of vaccine hesitancy. The study targets the COVID-19 vaccine specifically, and the COVID-19 vaccine could have a different set of attitudes and hesitancies as a result of its quick rollout and its unprecedented promotion campaign. As we understand it, the study did not collect any variables that related to regular vaccine hesitancy. Thus, we cannot differentiate the COVID-19 vaccine hesitant participants were hesitant to all vaccines or only COVID-19 vaccine specifically. Does that affect your interpretation of your results?

Response 2. Thank you for the valuable comment. In our study we measure vaccine confidence specific to COVID-19. We agree that vaccine hesitancy is specific to type of vaccines and context, and our results may not generalize to other type of vaccines or non-pandemic context. To remain transparent about such limitation, we added one statement in the **Limitation** section, with texts read: “Fourth, we only measured people’s COVID-19 vaccine confidence attitudes in our survey, therefore, conclusions of current study might not be able to generalize to other vaccination contexts.”

Methods- Line 471: Excluding participants with linguistic and cognitive problems could lead to bias since these populations usually delay vaccination and have lower vaccination rates. Limitations of excluding these people need to be discussed. Additionally, was the telephone survey done in Cantonese or Mandarin or both?

Response 3. Thank you very much for the comment. We excluded participants with linguistic and cognitive problems because our surveys were conducted over telephone, these participants might not be able to provide clear responses under such

data collection approach. We also checked the disability population in Hong Kong in 2020, there were only 3% of the population with linguistic or cognitive problems in the whole population³. We assumed that such exclusion criteria might not produce much impact on our current results. However, we agree that this is a limitation of our study. In the **Methods**, we have specified that: “Telephone interviews were conducted using Cantonese or Mandarin which covered over 92% of the population in Hong Kong⁴.” Excluding people who did not speak these two language was also a limitation of our study. We have discussed this limitations of these exclusion criteria in the **Limitation** section. The texts are: “Fifth, we excluded participants with linguistic and cognitive problems because these participants may not provide clear responses over a telephone interview. Such exclusion limits our understanding of vaccine hesitancy in these minority groups.”

Results- is it possible to provide a table showing vaccination coverage by different individual-level (i.e., demographic variables, trust in governments, etc.) and community-level attributes at different stages? This could let readers intuitively and statistically see changes.

Response 4. We appreciate this suggestion. Our study aims to map the multilevel determinants of vaccine uptake and their interconnections simultaneously throughout different stages of the COVID-19 vaccination programme. Therefore, providing the final vaccine uptake rates by each potential determinant in the main texts may confuse readers. However, we have added a supplementary table to provide further information on how the COVID-19 vaccine uptake rates varied by important determinants (demographics, married status, chronic status, and trust in government) in the four periods. Please find **Supplementary Table 1** in the Supplementary files for more information.

Results-Line 248-249. The network results show that “trust in government” is not directly associated with the outcome of vaccine uptake but through vaccine

confidence. However, in the qualitative part, “trust in government” is among the main reasons for COVID-19 vaccine hesitancy. Do you see this as an inconsistency?

Response 5. Thank you very much for this insightful comment. In the network models, we found that trust in government served as an important bridging node by linking other variables in the networks with COVID-19 vaccine confidence attitudes which in turn were directly and strongly associated with vaccine uptake. In the verbal reasons analysis, we found that participants tended to co-mention distrust in government with other reasons such as concern about vaccine safety. Therefore, the two parts of analyses are aligned with each other, with the network analyses providing more insights about the role of trust in government in shaping the whole network of determinants of vaccination uptake. We have included more discussions about the role of trust in government in shaping vaccination decision in the **Discussion**. The texts read: “Network results revealed that trust in government is an important bridging node that connected individual and interpersonal factors with vaccine-specific factors of confidence attitudes which in turn are directly and strongly associated with vaccination uptake. Specifically, individual demographics and perceived risk of COVID-19 and self-efficacy in preventing COVID-19 were first linked to trust in government which further connected to vaccine confidence attitudes. This indicates that the effects of individual factors on COVID-19 vaccine hesitancy and uptake could be partly because that trust in government varied by these individual factors. The analysis of the verbal reasons of vaccine hesitancy suggested that distrust in government was a frequently mentioned reason for refusing the primary vaccine doses uptake among the youngest people (aged 18-24). We also found that distrust in the government tended to be co-mentioned with other reasons. Both the network analysis and the analysis of the verbal reasons consistently indicate that trust in government is important to bridge different determinants of vaccination uptake, highlighting the importance of building trust for addressing vaccine hesitancy and improving vaccination uptake across individuals of different characteristics.”

Discussion- Line 364. Could you be more specific about vaccine risk? Were people worried about the side effects, whether the vaccine is not safe, or the vaccine is not effective enough to prevent them from COVID-19 infection?

Response 6. Thank you very much for the valuable comment. We revised the statement in the Discussion section to make it clearer to follow, with texts read: “Overall, this finding suggests that the non-pharmaceutical preventive measures remained important for people who would like to avoid any vaccine side effects or other adverse effects³⁹.”

Discussion- There is not enough discussion of the interaction of determinants, which is part of the aim of the study.

Response 7. Thank you very much for the valuable comment. We have included more discussions about the interconnections between determinants of vaccination uptake in the Discussion. The texts are: “Network results revealed that trust in government is an important bridging node that connected individual and interpersonal factors with vaccine-specific factors of confidence attitudes which in turn are directly and strongly associated with vaccination uptake. Specifically, individual demographics and perceived risk of COVID-19 and self-efficacy in preventing COVID-19 were first linked to trust in government which further connected to vaccine confidence attitudes. This indicates that the effects of individual factors on COVID-19 vaccine hesitancy and uptake could be partly because that trust in government varied by these individual factors. The analysis of the verbal reasons of vaccine hesitancy suggested that distrust in government was a frequently mentioned reason for refusing the primary vaccine doses uptake among the youngest people (aged 18-24). We also found that distrust in the government tended to be co-mentioned with other reasons. Both the network analysis and the analysis of the verbal reasons consistently indicate that trust in government is important to bridge different determinants of vaccination uptake,

highlighting the importance of building trust for addressing vaccine hesitancy and improving vaccination uptake across individuals of different characteristics.”

Thank you once again for providing all the valuable comments and advice for improving our manuscript.

Reviewer 3:

The manuscript focused the problem of identifying key determinants and their interactions associated with the primary and booster vaccination uptake at different stages of the COVID-19 vaccination campaign. The authors considered a set of variables of different types, including categorical, continuous and count data. The authors employed an existing R-package on mixed graphical model estimation, and an existing R-package using non-parametric bootstrap to provide confidence intervals of the edge weights for the estimated networks. The authors also provided their code for the network estimation and analysis. The analysis looks okay to me.

Response: Thank you for taking time to read our manuscript and review our analytical procedures. Your recognition is highly appreciated.

References

- 1 Larson, H. J., Jarrett, C., Eckersberger, E., Smith, D. M. & Paterson, P. Understanding vaccine hesitancy around vaccines and vaccination from a global perspective: a systematic review of published literature, 2007-2012. *Vaccine* **32**, 2150-2159 (2014).
- 2 Hong Kong Special Administrative Region. Hong Kong Vaccination Dashboard. <https://www.covidvaccine.gov.hk/en/dashboard/> (2023).
- 3 Hong Kong Equal Opportunities Commission. Equality for Diverse Abilities in Hong Kong. chrome-extension://efaidnbmnnnibpcajpcglclefindmkaj/https://www.eoc.org.hk/Upload/files/about-the-eoc/Diverse%20Ability%20Equality-Eng%20(Mar%202022).docx%20-%20Copy%201.pdf (2022).
- 4 Hong Kong Special Administrative Region. 2021 Population Census. <https://www.censtatd.gov.hk/en/scode600.html> (2021).

REVIEWERS' COMMENTS

Reviewer #1 (Remarks to the Author):

I congratulate the authors on producing this important piece of research. I think they have adequately addressed all of the reviewer comments.

Reviewer #2 (Remarks to the Author):

The authors responded appropriately to the reviewers. I note that both reviewer 1 and 2 did not have statistical expertise. If reviewer 3 provided that statistical expertise, I am okay with proceeding, otherwise a statistical expert in these methods should be found.